# Settling the Maximin Share Fairness for Scheduling among Groups of Machines

Bo Li [* 1]   Fangxiao Wang [* 1]   Shiji Xing [* 1]

## Abstract

We study the fair scheduling of jobs among groups of (unrelated) machines and focus on the maximin share (MMS) fairness at the group level. The problem was first introduced by Li et al. [NeurIPS 2023], where each group consists of a number of identical machines (or identical up to different speeds), and the cost of a group is determined by the minimum makespan on completing all jobs assigned to it. It is left as an open problem when the machines within each group are unrelated. In this paper, we first resolve this problem and design a polynomial-time algorithm that computes a 2-approximate MMS allocation via linear programming techniques. We complement this result with a hard instance, showing that no algorithm can be better than $(2 - \frac{1}{n})$-approximate MMS, where $n$ is the number of machines. Thus the approximation ratio 2 is asymptotically tight. When the groups consist of identical machines, we improve the approximation ratio to $\frac{4}{3}$.

## 1. Introduction

Fairness has long been a fundamental concern in social and economic environments. In the recent decade, it has drawn increasingly more attention from artificial intelligence, machine learning, and computer science due to the algorithmic and computation issues. One particularly noteworthy setting within the fairness discourse is resource allocation, which is not only theoretically intriguing but also holds practical importance. For an in-depth understanding of fair resource allocation, we recommend handbooks (Wooldridge, 2009; Brandt et al., 2016) and surveys (Moulin, 2019; Amanatidis et al., 2023) for a comprehensive overview.

---
[*]Equal contribution [1]Department of Computing, The Hong Kong Polytechnic University, Hong Kong, China. Correspondence to: Bo Li <comp-bo.li@polyu.edu.hk>, Fangxiao Wang <fangxiao.wang@connect.polyu.hk>, Shiji Xing <shiji.xing@connect.polyu.hk>.

*Proceedings of the 42^{nd} International Conference on Machine Learning*, Vancouver, Canada. PMLR 267, 2025. Copyright 2025 by the author(s).

In this paper, we focus on a job scheduling problem among groups of machines, which was first introduced in (Li et al., 2023). There are two layers of objects, where a number of indivisible jobs need to be first allocated to $n$ groups (viewed as super-agents). Each group controls a number of machines (which can be viewed as atomic agents). Upon receiving a set of jobs, the group should further assign these jobs to its machines with the objective of completing all assigned jobs as early as possible. Equivalently, we can define a cost function for each group that equals the minimum makespan of completing all jobs assigned to it. This setting is natural in practice; for instance, a group's workday can only conclude when all members have completed their assigned duties (e.g., facilitating the departure of a shuttle bus).

The fairness in this model is measured by the maximin share (MMS), introduced by Budish (2011). The MMS value of each group is the minimum cost it can ensure if the group partitions all jobs into $n$ bundles and always receives the bundle with maximum cost. Li et al. (2023) proved that if the machines in each group are identical (i.e., they take the same time to complete a task), a 2-approximate MMS allocation exists and can be found in polynomial time. Although Li et al. (2023) extended their results to related machines (i.e., the machines may have different speeds) within groups, the general setting of unrelated machines remains an open problem. In this work, we aim to resolve this open problem and also improve the approximation ratio for identical machines.

The above problem aligns two separate lines of research in the literature, namely MMS fair allocation of chores and job scheduling problem. The majority of research in MMS fair allocation assumes additive valuations, which is a special case of the setting in the current paper when each group consists of a single machine (Huang & Lu, 2021; Huang & Segal-Halevi, 2023). As proved in (Li et al., 2023), if agents have general subadditive valuations, MMS fairness does not admit a better than $n$ approximation. Our work presents a noteworthy class of valuations between additive and subadditive for which we can design (efficient) constant approximation algorithms.

On the other hand, the cost functions in our problem involve solving the optimization problem of scheduling among unrelated machines – a well-known NP-hard problem. The

best-known approximation is $2 - \frac{1}{m}$, $m$ being the number of jobs, as proved in (Shchepin & Vakhania, 2005), and Lenstra et al. (1990) showed that the problem does not admit better than $\frac{3}{2}$ polynomial-time approximation if P $\neq$ NP. The tight approximation ratio remains unknown; see Open Problem 4 in (Schuurman & Woeginger, 1999). It is worth noting that finding approximate MMS fair allocations itself does not rely on the assumption of P vs NP, where the impossibility is universal and obtained from hard instances. Our work generalizes this model to scheduling among *groups* of unrelated machines.

Our results are summarized in the following section.

### 1.1. Our Results

We consider the fair allocation problem among $n$ asymmetric groups, where each group $i$ consists of $g_i$ (unrelated) private machines who are referred to as agents. Each machine has an additive cost function (i.e., processing time) over the jobs. The maximin share (MMS) (Budish, 2011) of each group is defined as the smallest cost it can guarantee if it divides the jobs into $n$ bundles but receives the bundle with the largest cost. The cost of a bundle of jobs for a group is the minimum makespan of completing these jobs, i.e., the minimum of the longest processing time by scheduling the jobs in this bundle to the machines within the group. To indicate the further distribution of the jobs among machines/agents within each group, we explicitly call this share *group MMS*, abbreviated as GMMS. An allocation is $\alpha$-approximate GMMS fair ($\alpha$-GMMS) to a group if its cost is no greater than $\alpha$ times its GMMS cost. Our objective is to understand to what extent we can allocate the jobs to groups so that GMMS fairness is (approximately) satisfied.

We distinguish between two cases: if the machines within each group are unrelated, we call it the *heterogeneous* setting; if they are identical, we call it the *homogeneous* setting.

Our results can be summarized as follows.

**Result 1** A 2-GMMS fair allocation can be computed in polynomial time in the heterogeneous setting.

Our algorithm to compute 2-GMMS allocations relies on LP rounding. We begin by constructing a fractional GMMS allocation in which a machine is assigned a positive fraction of a job only if the job's cost does not exceed the GMMS of the machine's group. Our method generalizes the LP techniques developed in (Lenstra et al., 1990) to accommodate multiple groups. Then, we give a rounding procedure that converts the fractional allocation into an integral one, where each machine will have at most one fractional item in their bundle rounded up to integral, resulting in a 2-GMMS allocation. To complement this result, we show that no algorithm can be better than $(2 - \frac{1}{n})$-GMMS.

**Result 2** For any $n \geq 2$, there is an instance with $n$ groups such that no allocation is better than $(2 - \frac{1}{n})$-GMMS.

Finally, we improve the approximation ratio to $\frac{4}{3}$ in the homogeneous setting, which improves the approximation of 2 proved by Li et al. (2023).[1]

**Result 3** A $\frac{4}{3}$-GMMS fair allocation exists in the homogeneous setting.

Generalizing the hard instance in (Kurokawa et al., 2018), we can show that for any number of agents, exact GMMS allocations may not exist, which is true even when each group contains a single agent. In fact, in this case, the problem degenerates to the case of chore allocation with additive cost functions and thus does not admit better than $\frac{44}{43}$ approximation when $n = 3$ (Feige et al., 2021).

### 1.2. Related Works

The MMS fair division has been widely studied in the past years, with a particular focus on additive functions. For allocating goods, Kurokawa et al. (2018) first observed instances which do not admit MMS allocations, and proved the existence of $\frac{2}{3}$-MMS allocations. Amanatidis et al. (2017) designed a polynomial time algorithm with same approximation ratio, which was later improved to $\frac{3}{4}$ by Ghodsi et al. (2018). Garg & Taki (2021), Akrami et al. (2023a), Akrami & Garg (2024) further improved the approximation ratio to slightly larger than $\frac{3}{4}$. Feige et al. (2021) proved that no algorithm has an approximation ratio better than $\frac{39}{40}$. Regarding chores allocation, it is known that no algorithm can guarantee an approximation ratio better than $\frac{44}{43}$ (Feige et al., 2021), and constant approximation algorithms are given in (Aziz et al., 2017; Huang & Lu, 2021; Huang & Segal-Halevi, 2023).

In many real-world cases, the functions are more complicated. For instance, the functions appear in machine learning and artificial intelligence are often submodular (Bilmes, 2022). For goods, to approximate the MMS allocation, Barman & Krishnamurthy (2020); Ghodsi et al. (2022); Uziahu & Feige (2023) and (Akrami et al., 2023b; Seddighin & Seddighin, 2024) respectively gave constant approximation algorithms for submodular and XOS valuations. Ghodsi et al. (2022); Seddighin & Seddighin (2024); Feige & Huang (2025) proved the existence of logarithmic approximations. For chores allocation, Li et al. (2023) proved a lower bound of $\min\{n, \frac{\log m}{\log \log m}\}$ for submodular cost functions with binary margins. They also proved the existence of $\min\{n, \log m\}$-approximate MMS allocations, and improved the approximations to constants in two special

---

[1] Li et al. (2023) extended their result to the setting of related machines, where machines within a group have different speeds, i.e., the processing times are identical up to different scales.

models, where the cost functions involve solving the bin packing and job scheduling problems. Recently, Wang & Li (2024) considered the vertex cover model and also proved the existence of constant approximate MMS fair allocations. In another line, Barman et al. (2023) proved the existence of an exact MMS fair and Pareto optimal allocation under binary supermodular cost functions.

## 2. Preliminaries

For any integer $k \geq 1$, let $[k] = \{1, \ldots, k\}$. In a group fair allocation instance, there are $n$ groups and $m$ indivisible chores, denoted by $\mathcal{G} = \{G_1, \ldots, G_n\}$ and $\mathcal{M} = \{e_1, \ldots, e_m\}$. Each group $G_i$ contains $g_i$ machines which are referred to as agents, i.e., $G_i = \{a_{i,1}, \ldots, a_{i,g_i}\}$ and $g_i = |G_i| \geq 1$. It is assumed that the groups do not overlap, i.e., $G_i \cap G_j = \emptyset$ for all $i \neq j$. Denote by $g = \sum_{i \in n} g_i$ the total number of agents. Each individual agent $a_{i,j}$ has a cost function $c_{i,j} : 2^{\mathcal{M}} \to \mathbb{R}_{\geq 0}$, where $c_{i,j}(S)$ represents her cost (or finishing time) on completing the chores in $S \subseteq \mathcal{M}$. For simplicity, denote $c_{i,j}(e) = c_{i,j}(\{e\})$. In this paper, the cost functions are assumed to be additive, i.e., $c_{i,j}(S) = \sum_{e \in S} c_{i,j}(e)$. Let $\mathbf{c}_i = (c_{i,1}, \ldots, c_{i,g_i})$ and $\mathbf{c} = (\mathbf{c}_1, \ldots, \mathbf{c}_n)$. We distinguish two cases. When agents in a group have the same cost function, i.e., $c_{i,j}(e) = c_{i,l}(e)$ for all $G_i \in \mathcal{G}$, $e \in \mathcal{M}$ and $a_{i,j}, a_{i,l} \in G_i$, we call it a *homogeneous* setting, and otherwise, a *heterogeneous* setting.

An allocation is denoted by $\mathbf{A} = (\mathbf{A}_1, \ldots, \mathbf{A}_n)$, where $\mathbf{A}_i = (A_{i,1}, \ldots, A_{i,g_i})$ is the allocation to group $G_i$ and $A_{i,j}$ is the allocation to agent $a_{i,j}$. When there is no confusion, we also denote $\mathbf{A}_i = A_{i,1} \cup \cdots \cup A_{i,g_i}$. It is required that all items are allocated and every item is allocated exactly once, i.e, $\bigcup_{i \in [n]} \bigcup_{j \in [g_i]} A_{i,j} = \mathcal{M}$ and $A_{i_1,l_1} \cap A_{i_2,l_2} = \emptyset$ for $(i_1, l_1) \neq (i_2, l_2)$. Finally, we use $\mathcal{I} = (\mathcal{G}, \mathcal{M}, \mathbf{c})$ to denote a group chore allocation instance.

The agents in the same group collaborate with each other and have reached an agreement to complete their assigned items together so as to minimize the maximum cost. Therefore, they share the same fairness benchmark value which depends on the largest individual cost. To formally define the group maximin share, we first recall the original definition of maximin share (MMS). For any integer $k \geq 1$ and set $S \subseteq \mathcal{M}$, denote by $\Pi(S, k)$ the set of all $k$-partitions of $S$. When allocating items $S$ among $k$ agents, the MMS of an agent with cost function $c$ is defined as

$$\mathsf{MMS}(S, k, c) = \min_{(S_1, \ldots, S_k) \in \Pi(S, k)} \max_{1 \leq j \leq k} c(S_j).$$

The *group maximin share* (GMMS) of group $G_i$ is the optimal worst-case individual share by partitioning the items into $g_i$ bundles where each agent $a_{i,k}$ shares responsibility of her bundle with the other $n - 1$ groups. That is

$$\mathsf{GMMS}_i = \min_{(S_1, \ldots, S_{g_i}) \in \Pi(\mathcal{M}, g_i)} \max_{1 \leq k \leq g_i} \mathsf{MMS}(S_k, n, c_{i,k}).$$

$$(1)$$

If a partition $(S_1^*, \ldots, S_{g_i}^*)$ of $\mathcal{M}$ reaches the value of $\mathsf{GMMS}_i$, i.e.,

$$\max_{1 \leq k \leq g_i} \mathsf{MMS}(S_k^*, n, c_{i,k}) = \mathsf{GMMS}_i,$$

it is called a GMMS-defining partition to group $G_i$.

**Definition 2.1** ($\alpha$-GMMS). Given $\alpha \geq 1$, an allocation $\mathbf{A}$ is called $\alpha$-approximate GMMS fair ($\alpha$-GMMS), if

$$c_{i,j}(A_{i,j}) \leq \alpha \cdot \mathsf{GMMS}_i, \text{ for all } j \in [g_i] \text{ and } i \in [n].$$

We have the following simple bound for $\mathsf{GMMS}_i$.

**Lemma 2.2.** $\mathsf{GMMS}_i \geq \frac{1}{n \cdot g_i} \sum_{e \in \mathcal{M}} \min_{1 \leq k \leq g_i} c_{i,k}(e)$.

Alternatively, we can introduce a cost function for each group to measure the collaborative cost of the agents in the group, as in (Li et al., 2023). This is partly motivated by the job scheduling problem, where the makespan decides the overall cost of the group so that the agents in the group share the same "actual" cost. Formally, the cost function of group $G_i$ is denoted by $C_i : 2^{\mathcal{M}} \to \mathbb{R}_{\geq 0}$ where for all $S \subseteq \mathcal{M}$,

$$C_i(S) = \min_{(S_1, \ldots, S_{g_i}) \in \Pi(S, g_i)} \max_{1 \leq j \leq g_i} c_{i,j}(S_j).$$

Then we have the following lemma, which presents an equivalent definition of GMMS.

**Lemma 2.3.**

$$\mathsf{GMMS}_i = \min_{(S_1, \ldots, S_n) \in \Pi(\mathcal{M}, n)} \max_{1 \leq j \leq n} C_i(S_j). \quad (2)$$

Note that the group cost functions are subadditive, i.e., $C(S_1 \cup S_2) \leq C(S_1) + C(S_2)$ for any $S_1$ and $S_2$. In fact, the model in (Li et al., 2023) used Lemma 2.3 to study the MMS allocation of subadditive cost functions. By their results, we have an $O(\min\{n, \log m\})$ approximation. If agents in each group have identical cost functions, the approximation ratio improves to 2, but it's unknown if a better approximation exists for the general heterogeneous case.

Our paper examines two case types: Section 3 covers cases with *heterogeneous agents*, where agents in a group have varied cost functions, while Section 4 addresses cases with *homogeneous agents* where agents within a group share identical cost functions.

## 3. Heterogeneous Agents

In this section, we present our main result. We first prove that no algorithm can perform better than $(2 - \frac{1}{n})$-GMMS by constructing a hard instance and then design a polynomial-time algorithm that always computes a 2-GMMS allocation for all instances.

## 3.1. Lower Bound

**Theorem 3.1.** *There is no algorithm that can perform better than* $(2 - \frac{1}{n})$*-GMMS, where $n$ is the number of groups.*

*Proof.* We construct the following instance with $n$ groups $\{G_1, \ldots, G_n\}$, $2n - 1 + n(n-1)^2$ agents and $n + n^2(n-1)$ items. The items are divided into two types:

- $Q = \{e_l \mid l \in [n]\}$;

- $T = \{e_{l,p,k} \mid l \in [n], p \in [n-1], \text{ and } k \in [n]\}$,

where the items in $T$ are ordered by their indices, as shown in Figure 1. Note that $|Q| = n$ and $|T| = n^2(n-1)$. We consider the following two different partitions of $T$, $T = \bigcup_{p \in [n-1], k \in [n]} T_{p,k}$ and $T = \bigcup_{k \in [n]} T_k$, where

$$T_{p,k} = \{e_{l,p,k} \in T \mid 1 \le l \le n\}$$

and

$$T_k = \{e_{l,p,k} \in T \mid 1 \le l \le n \text{ and } 1 \le p \le n-1\}.$$

The above partitions of $T$ are illustrated in Figure 1. The green items correspond to one $T_{p,k}$ (in fact, it is $T_{n-1,1}$) and the orange items correspond to one $T_k$ (in fact, it is $T_4$).

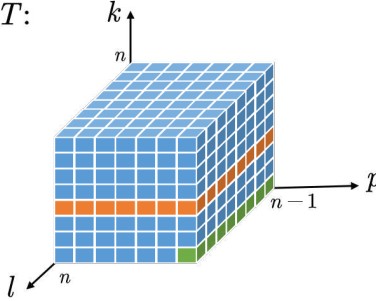

Figure 1. The structure of items in $T$, where $T_{n-1,1}$ is marked in green and $T_4$ is marked in orange.

We next define the cost functions of the agents.

For $i = 1, \ldots, n-1$, group $G_i$ contains $(n-1)n + 1$ agents denoted by $a_{i,1}$ and $a_{i,(p,k)}$ with $p \in [n-1]$ and $k \in [n]$.

- For agent $a_{i,1}$,
  - $c_{i,1}(e_l) = 1$ for all $e_l \in Q$,
  - $c_{i,1}(e_{l,p,k}) = \infty$ for all $e_{l,p,k} \in T$.

- For each agent $a_{i,(p,k)}$ where $p \in [n-1]$ and $k \in [n]$,
  - $c_{i,(p,k)}(e_{l,p,k}) = 1$ for all $e_{l,p,k} \in T_{p,k}$,
  - $c_{i,(p,k)}(e_{l,p',k'}) = \infty$ for all $e_{l,p',k'} \in T_{p',k'}$ with $p' \ne p$ or $k' \ne k$,

- $c_{i,(p,k)}(e_l) = \infty$ for all $e_l \in Q$.

*Claim 1.* $\mathsf{GMMS}_i = 1$ for $i = 1, \ldots, n-1$.

*Proof.* It is easy to see that $\mathsf{GMMS}_i \ge 1$ since the smallest single-item cost of every agent is 1. Consider partition $(B_1, \ldots, B_n)$ with $B_l = \{e_l\} \cup \{e_{l,p,k} \in T_{p,k} \mid p \in [n-1] \text{ and } k \in [n]\}$. It can be verified that $C_i(B_l) = 1$ by allocating $e_l$ to $a_{i,1}$ and $e_{l,(p,k)}$ to $a_{i,(p,k)}$, and thus $c_{i,1}(e_l) = c_{i,(p,k)}(e_{l,(p,k)}) = 1$. □

Group $G_n$ contains $n$ agents $\{a_{n,1}, \ldots, a_{n,n}\}$. For each agent $a_{n,k}$, $1 \le k \le n$,

- $c_{n,k}(e_k) = 1$ for each $e_k \in Q$,

- $c_{n,k}(e_l) = \infty$ for each $e_l \in Q$ with $l \ne k$,

- $c_{n,k}(e_{l,p,k}) = \frac{1}{n}$ for each $e_{l,p,k} \in T_k$,

- $c_{n,k}(e_{l,p,k'}) = \infty$ for each $e_{l,p,k'} \in T_{k'}$ with $k' \ne k$.

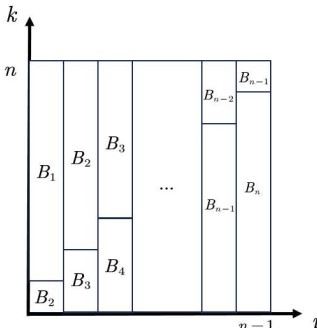

Figure 2. Partition $(B_1, \ldots, B_n)$, where the illustration is Figure 1 projected on the $l$-axis. That is, each item $(p, k)$ in Figure 2 represents a row of $n$ items $(p, k, l)$ with $l = 1, \ldots, n$.

*Claim 2.* $\mathsf{GMMS}_n = 1$.

*Proof.* By Lemma 2.2,

$$\mathsf{GMMS}_n \ge \frac{1}{n^2} \sum_{e \in \mathcal{M}} \min_{1 \le k \le n} c_{n,k}(e)$$

$$= \frac{1}{n^2} \sum_{1 \le k \le n} \left( c_{n,k}(e_k) + \sum_{l \in [n], p \in [n-1]} c_{n,k}(e_{l,p,k}) \right) = 1.$$

Consider partition $(B_1, \ldots, B_n)$ with

$$B_q = \{e_q\} \cup \{e_{l,q-1,k} \mid k < q, 1 \le l \le n\}$$
$$\cup \{e_{l,q,k} \mid k > q, 1 \le l \le n\},$$

where $\{e_{l,0,k} \mid 1 \le k < 1, l \le n\} = \emptyset$ and $\{e_{l,n,k} \mid 1 \le k > n, l \le n\} = \emptyset$, as shown in Figure 2. That is, bundle

$B_q$ obtains $n$ items on each column $k \neq q$ and nothing on column $k = q$. To see group $G_n$'s cost is 1 for $B_q$, we allocate $e_q$ to $a_{n,q}$, so $a_{n,q}$'s cost is 1; allocate $\{e_{l,q-1,k} \mid 1 \leq l \leq n\}$ to $a_{n,k}$ for $k < q$, and $\{e_{l,q,k} \mid 1 \leq l \leq n\}$ to $a_{n,k}$ for $k > q$, so that every agent $a_{n,k}$ for $k \neq q$ has $n$ items, each of which has cost $\frac{1}{n}$. $\square$

Next we prove that for any allocation $\mathbf{A} = (\mathbf{A}_1, \ldots, \mathbf{A}_n)$, at least one of the groups has a makespan of at least $2 - \frac{1}{n}$. We start with two simple cases.

Case 1. If $|\mathbf{A}_i \cap Q| \geq 2$ for any group $G_i$, $1 \leq i \leq n-1$, then $C_i(\mathbf{A}_i) \geq 2$. This is because only agent $a_{i,1}$ has cost 1 on items in $Q$ while all other agents have cost infinity.

Case 2. If $|\mathbf{A}_i \cap T_{p,k}| \geq 2$ for any group $G_i$, $1 \leq i \leq n-1$, $1 \leq p \leq n-1$ and $1 \leq k \leq n$, then $C_i(\mathbf{A}_i) \geq 2$. This is because only agent $a_{i,(p,k)}$ has cost 1 on items in $T_{p,k}$ while all other agents have cost infinity.

Case 3. If $|\mathbf{A}_n \cap Q| \geq 1$ and $|\mathbf{A}_i \cap T_{p,k}| \leq 1$ for all $1 \leq i \leq n-1$ and $1 \leq p \leq n-1$ and $1 \leq k \leq n$, we claim that $C_n(\mathbf{A}_n) \geq 2 - \frac{1}{n}$. Suppose $e_{k^*} \in A_n \cap Q$, and thus $e_{k^*}$ must be allocated to agent $a_{n,k^*}$. We next consider the allocation of $T_{k^*}$. Since $|\mathbf{A}_i \cap T_{p,k^*}| \leq 1$ for all $1 \leq i \leq n-1$ and $1 \leq p \leq n-1$, we have $|\mathbf{A}_i \cap T_{k^*}| \leq n-1$, and thus

$$\sum_{i=1}^{n} |\mathbf{A}_i \cap T_{k^*}| \leq (n-1)^2.$$

That is

$$|\mathbf{A}_n \cap T_{k^*}| \geq n(n-1) - (n-1)^2 = n-1.$$

Since $c_{n,k}(e_{l,p,k^*}) = \infty$ for all $k \neq k^*$, all items in $A_n \cap T_{k^*}$ are allocated to agent $a_{n,k^*}$. Therefore

$$C_n(\mathbf{A}_n) \geq c_{n,k^*}(e_{k^*}) + c_{n,k^*}(A_n \cap T_{k^*})$$
$$1 + \frac{1}{n}(n-1) = 2 - \frac{1}{n}.$$

Combining Cases 1, 2 and 3 completes the proof. $\square$

### 3.2. Upper Bound: A 2-GMMS Algorithm

**Theorem 3.2.** *A 2-GMMS allocation can be computed in polynomial time for any instance.*

Before showing our algorithm, we first introduce additional definitions. We use $\mathbf{x} = (\mathbf{x}_1, \ldots, \mathbf{x}_n)$ to denote a fractional allocation, where $\mathbf{x}_i = (\mathbf{x}_{i,1}, \ldots, \mathbf{x}_{j,g_i})$ and $\mathbf{x}_{i,j} = (x^1_{i,j}, \ldots, x^m_{i,j})$ such that each $0 \leq x^l_{i,j} \leq 1$ denotes the fraction of item $e_l \in \mathcal{M}$ that agent $a_{i,j}$ obtains. Thus, a feasible allocation satisfies $\sum_{i\in[n],j\in[g_i]} x^l_{i,j} = 1$ for all $1 \leq l \leq m$. For simplicity, let $c_{i,j}(\mathbf{x}_{i,j}) = \sum_{l\in[m]} c_{i,j}(e_l) \cdot x^l_{i,j}$.

**Definition 3.3.** For a fractional allocation $\mathbf{x}$, we construct an auxiliary (multi-)graph $G(\mathbf{x}) = (V, E)$, where the vertices $V = G_1 \cup \cdots \cup G_n$ represent the agents. If $x^l_{i_1,j_1} \neq 0$ and $x^l_{i_2,j_2} \neq 0$, we construct an edge between $a_{i_1,j_1}$ and $a_{i_2,j_2}$ and label the edge by $l$. A cycle in the graph is called *share cycle* if at least two edges in the cycle have different labels.

**Lemma 3.4.** *There exists a fractional allocation* $\mathbf{x}$ *such that*

- *For every agent* $a_{i,j}$, $c_{i,j}(\mathbf{x}_{i,j}) \leq \mathsf{GMMS}_i$.

- *For any* $x^l_{i,j} \neq 0$, $c_{i,j}(e_l) \leq \mathsf{GMMS}_i$.

- *There is no share-cycle in the auxiliary graph.*

*Further, such an allocation can be found in polynomial time.*

*Proof.* We utilize the linear programming technique introduced in (Lenstra et al., 1990) to prove this lemma.

Fix a parameter $t_i \geq 0$, which can be viewed as a guess of the value of $\mathsf{GMMS}_i$. Let

$$\mathcal{M}_{i,j}(t_i) = \{e \in \mathcal{M} \mid c_{i,j}(e) \leq t_i\}$$

be the set of items for which agent $a_{i,j}$ has cost no greater than $t_i$, and

$$P_{i,l}(t_i) = \{a_{i,j} \in G_i \mid c_{i,j}(e) \leq t_i\}$$

be the set of agents in $G_i$ who have cost on item $e_l$ no greater than $t_i$. Consider the following linear program $LP_i(t_i)$:

$$\begin{cases} \sum_{e_l \in \mathcal{M}_{i,j}(t_i)} c_{i,j}(e_l) \cdot y^l_{i,j,k} \leq t_i, & \forall j \in [g_i], k \in [n] \\ \sum_{j \in P_{i,l}(t_i), k \in [n]} y^l_{i,j,k} = 1, & \forall e_l \in \mathcal{M} \\ y^l_{i,j,k} \geq 0, & \forall j \in [g_i], k \in [n], e_l \in \mathcal{M}_{i,j}(t_i), \end{cases}$$

where $i, j, k, l$ represent group $G_i$, agent $a_{i,j}$ in $G_i$, the $k$-th bundle in the $n$-partition of group $G_i$, and item $e^l$. Thus, variable $y^l_{i,j,k}$ means the fraction of item $e^l$ allocated to agent $a_{i,j}$ in the $k$-th bundle in the $n$-partition of group $G_i$. The first constraint means that the cost of every copy of every agent is not greater than $t_i$, and the second constraint means all items are allocated.

Let $t^*_i$ be the minimum value of $t_i$ such that $LP(t_i)$ has a feasible solution. Then we have the following claims.

*Claim* 3. $t^*_i$ *can be computed in polynomial time.*

*Proof.* Similar to (Lenstra et al., 1990), we can binary search $t^*_i$. An upper bound of $t^*_i$ is the maximum cost among all agents in $G_i$ by allocating each item to the agent with minimum cost on it, and a lower bound is $\frac{1}{g_i \cdot n}$ times this maximum cost. For each $t_i$, checking whether $LP(t_i)$ admits a feasible solution can be done in polynomial time. $\square$

*Claim* 4. $t_i^* \leq \mathsf{GMMS}_i$.

*Proof.* This lemma is straightforward as $LP_i(\mathsf{GMMS}_i)$ has a feasible (integral) solution. □

We next construct a fractional allocation $\mathbf{x} = (\mathbf{x}_1, \ldots, \mathbf{x}_n)$ satisfying all conditions of Lemma 3.4. Denote by $\{y_{i,k,j}^{l*}\}$ a feasible solution of $LP_i(t_i^*)$. Let

$$x_{i,j}^l = \frac{1}{n} \sum_{k \in [n]} y_{i,k,j}^{l*}.$$

By the first constraint of $LP_i(t_i^*)$,

$$c_{i,j}(\mathbf{x}_{i,j}) \leq \frac{1}{n} \cdot n \cdot t_i^* \leq \mathsf{GMMS}_i$$

Moreover, $x_{i,j}^l > 0$ if and only if $y_{i,k,j}^{l*} > 0$ for some $k \in [n]$. By the definition of $\mathcal{M}_{i,j}(t_i), c_{i,j}(e_l) \leq t_i^* \leq \mathsf{GMMS}_i$. Thus, $\mathbf{x}$ satisfies the first two conditions of Lemma 3.4. If there is no share cycle in the auxiliary graph $G(\mathbf{x})$, the lemma holds. Otherwise, we gradually modify $\mathbf{x}$ to satisfy the third condition without hurting the first two.

Let $\mathcal{C} = a_{i_1,j_1} - a_{i_2,j_2} - \cdots - a_{i_k,j_k} - a_{i_1,j_1}$ be a share cycle in the auxiliary graph $G(\mathbf{x})$. Denote $a_{i_1,j_1} \rightarrow a_{i_2,j_2} \rightarrow \cdots \rightarrow a_{i_k,j_k} \rightarrow a_{i_1,j_1}$ as the right direction and the backward direction as the left. Let item $e_l$ denote the item that agent $a_{i_l,j_l}$ and her right neighbor $a_{i_{l+1},j_{l+1}}$ share in the share cycle, where $l + 1 = 1$ if $l = k$. For agent $a_{i_l,j_l}$, denote by $\alpha_l = c_{i_l,j_l}(e_l)$ the cost of her right shared item of the cycle and $\beta_l = c_{i_l,j_l}(e_{l-1})$ as the cost of the left shared item, where $l - 1 = k$ if $l = 1$.

Note that either of the following expressions must be less than or equal to 1: $\frac{\prod_{i \in [l]} \alpha_i}{\prod_{i \in [l]} \beta_i}$ or $\frac{\prod_{i \in [l]} \beta_i}{\prod_{i \in [l]} \alpha_i}$, since their product is exactly 1. If $\frac{\prod_{i \in [l]} \alpha_i}{\prod_{i \in [l]} \beta_i} \leq 1$, we choose left as the desired direction and right otherwise. Without loss of generality, assume that the direction is right.

Let $\gamma > 0$ be a sufficiently small number. As Figure 3 shows, we let agent $a_{i_1,j_1}$ give a fraction of item $e_1$ with cost $\gamma$ to $a_{i_2,j_2}$. The amount of the passed item $e_1$ is $\gamma \cdot \frac{1}{\alpha_1}$. For agent $a_{i_2,j_2}$, the cost of the received fraction is $\gamma \cdot \frac{\beta_2}{\alpha_1}$. We want $a_{i_2,j_2}$ to pass a fraction to her right neighbor so that her total cost does not change. Therefore she will pass $\gamma \cdot \frac{\beta_2}{\alpha_1 \cdot \alpha_2}$ fraction of item $e_2$ to $a_{i_3,j_3}$. Repeat this process for the remaining agents in the cycle. For agent $a_{i_l,j_l}$ with $l \in [2, \ldots, k]$, she will receive a fraction of $e_{l-1}$ with cost $\gamma \cdot \frac{\prod_{i \in \{2,\cdots,l\}} \beta_i}{\prod_{i \in [l-1]} \alpha_i}$, and will pass a fraction $e_l$ with amount $\gamma \cdot \frac{\prod_{i \in \{2,\cdots,l\}} \beta_i}{\prod_{i \in [l]} \alpha_i}$ to her right neighbor. Eventually, agent $a_{i_1,j_1}$ receives a fraction of the item $e_k$ with cost $\gamma \cdot \frac{\prod_{i \in \{k\}} \beta_i}{\prod_{i \in [k]} \alpha_i}$, which is no greater than $\gamma$, the cost she gives

away. Therefore, her total cost does not increase. Hence, by exchanging items in the above way, no agent's cost is increased.

It remains to decide the value of $\gamma$ so that at least one edge in this cycle is canceled and no agent passes more items than she possesses. Consider agent $a_{i_l,j_l}$. If $e_l$ and $e_{l-1}$ are the same item, then her allocation $\mathbf{x}_{i_l,j_l}$ does not change in the process, since she essentially passes a fractional item from her left neighbor to her right. Therefore, the edges between $a_{i_l,j_l}$ and $a_{i_{l+1},j_{l+1}}$ cannot be canceled and she will not give more item than she possesses no matter how large $\gamma$ is. Thus we can safely exclude her from the calculation of $\gamma$. Consequently, we only need to focus on agents who share different items with their neighbors. Assume $a_{i_l,j_l}$ is such an agent. If $l = 1$, then she will pass a fraction of $e_1$ with amount $\gamma \cdot \frac{1}{\alpha_1}$. Otherwise, she will receive cost $\gamma \cdot \frac{\prod_{i \in \{2,\cdots,l\}} \beta_i}{\prod_{i \in [l-1]} \alpha_i}$, and needs to pass a fraction of $e_l$ with amount $\gamma \cdot \frac{\prod_{i \in \{2,\cdots,l\}} \beta_i}{\prod_{i \in [l]} \alpha_i}$. We need to select $\gamma$ such that $\gamma \cdot \frac{\prod_{i \in \{2,\cdots,l\}} \beta_i}{\prod_{i \in [l]} \alpha_i} \leq x_{i_l,j_l}^l$. Furthermore, since we want to remove one edge from the cycle, we also hope that this inequality holds tight for some agent. We then can determine the value $\gamma$ by the following:

$$\gamma = \min_{l \in S} \left\{ x_{i_l,j_l}^l \cdot \frac{\beta_l \cdot \prod_{i \in [l]} \alpha_i}{\prod_{i \in [l]} \beta_i} \right\},$$

where $S$ is the set of agents in the shared cycle who share different items with their neighbors. As a special case,

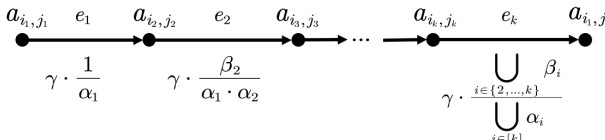

*Figure 3.* Exchanging items among the cycle in the right direction.

By reallocating the items in the cycle with the computed $\gamma$, it is ensured that at least one edge in this cycle is cancelled and the reallocation is valid. Therefore, after the reallocation, there is one less edge in the graph $G(\mathbf{x})$. Consequently, in polynomial time, we can repeat this process until all share cycles are eliminated. In the process, the total cost of agents are non-increasing, so $c_{i,j}(\mathbf{x}_{i,j}) \leq \mathsf{GMMS}_i$.

Furthermore, no agent receives a new item in the exchanging items process. For item $e_p$ and agent $a_{i,j}$ with $x_{i,j,p} > 0, c_{i,j}(e_p) \leq \mathsf{GMMS}_i$ still holds. □

Now we are ready to prove Theorem 3.2.

*Proof of Theorem 3.2.* We first obtain a fractional allocation $\mathbf{x}$ satisfying all conditions of Lemma 3.4.

If there is no share cycle in $G(\mathbf{x})$, then there exists an agent whose bundle contains only one fractional item. Assume not, then each agent who possesses fractional items must have at least two or more of them. Therefore, such agents will have at lease two adjacent edges with different labels in the auxiliary graph. Furthermore, since each non-isolated vertex in the auxiliary graph has a degree of at least 2, a share cycle must exist, making a contradiction.

Let $a_{i,j}$ be such an agent having received only one fractional item $e_p$. Let $A_{i,j} = \{e_l | x_{i,j}^l = 1\} \cup \{e_p\}$. Note that for agent $a_{i,j}$, $c_{i,j}(A_{i,j}) \leq c_{i,j}(\{e_l | x_{i,j}^l = 1\}) + c_{i,j}(\{e_p\}) \leq c_{i,j}(\mathbf{x}_{i,j}) + \mathsf{GMMS}_i \leq 2\mathsf{GMMS}_i$. Then, delete the agent $a_{i,j}$ and all the edges labeled with $e_p$ in $G(\mathbf{x})$.

Since the updated auxiliary graph $G(\mathbf{x})$ is a subgraph of the previous one, there is still no share cycle. So we keep the above process until there is no agent in $G(\mathbf{x})$ and we get a 2-GMMS allocation $\mathbf{A} = A_{i,j\,i\in[n],j\in[g_i]}$. $\qquad\square$

## 4. Homogeneous Agents

In this section, we switch our focus to the homogeneous setting where the agents within each group $G_i$ have identical cost functions. That is, for any two agents $a_{i,j}$, $a_{i,j'}$, $c_{i,j}(e) = c_{i,j'}(e)$ for all items $e \in \mathcal{M}$. For simplicity, we use $c_i(\cdot)$ to denote the cost function of all agents in group $G_i$. Li et al. (2023) proved that the homogeneous setting admits a 2-GMMS allocation. In the following, we improve this approximation ratio to $\frac{4}{3}$.

### 4.1. Upper Bound: A $\frac{4}{3}$-GMMS Algorithm

**Theorem 4.1.** *There exists a $\frac{4}{3}$-GMMS allocation for all homogeneous instances.*

**Definition 4.2** (Ordered Instance). *A homogeneous instance $\mathcal{I} = (\mathcal{G}, \mathcal{M}, \mathbf{c})$ is called* ordered *if the agents in all groups have the same ordinal preference over all items, i.e., $c_i(e_1) \geq c_i(e_2) \geq \cdots \geq c_i(e_m)$ for all $i \in [n]$.*

It is widely known that when each group contains exactly one agent, the ordered instance is the most challenging case for approximating GMMS fairness (Barman & Krishnamurthy, 2020; Huang & Lu, 2021; Li et al., 2022). This property also holds for the case where each group contains multiple identical agents (Li et al., 2023).

**Lemma 4.3.** *(Li et al., 2023) For any $\alpha \geq 1$, if an $\alpha$-GMMS allocation exists for all ordered instances with homogeneous agents, then an $\alpha$-GMMS allocation exists for all instances with homogeneous agents.*

By Lemma 4.3, it suffices to restrict our focus on ordered instances. In addition, we assume without loss of generality that the valuations of agents are normalized so that $c_i(\mathcal{M}) = n$ for all $i \in [n]$. By the definition of $\mathsf{GMMS}_i$, $n \cdot g_i \cdot$

$\mathsf{GMMS}_i \geq c_i(\mathcal{M}) = n$ and thus $g_i \cdot \mathsf{GMMS}_i \geq 1$.

We next introduce some notions to be used in the algorithm.

Recall that $c_i(e) \leq \mathsf{GMMS}_i, \forall i \in [n], e \in \mathcal{M}$. An item $e \in \mathcal{M}$ is called *large* for group $G_i$ if $c_i(e) > \frac{2}{3}\mathsf{GMMS}_i$, *medium* if $\frac{1}{3}\mathsf{GMMS}_i \leq c_i(e) \leq \frac{2}{3}\mathsf{GMMS}_i$, and *small* if $c_i(e) < \frac{1}{3}\mathsf{GMMS}_i$. For a set of chores $S$, let $H_i(S)$, $D_i(S)$ and $L_i(S)$ respectively denote the set of large, medium and small items in $S$ for group $G_i$, i.e.,

$$H_i(S) = \left\{ e_k \in S \,\Big|\, c_i(e_k) > \frac{2}{3}\mathsf{GMMS}_i \right\},$$

$$D_i(S) = \left\{ e_k \in S \,\Big|\, \frac{1}{3}\mathsf{GMMS}_i \leq c_i(e_k) \leq \frac{2}{3}\mathsf{GMMS}_i \right\},$$

$$L_i(S) = \left\{ e_k \in S \,\Big|\, c_i(e_k) < \frac{1}{3}\mathsf{GMMS}_i \right\}.$$

Let $(S_1, \ldots, S_{g_i})$ be an arbitrary GMMS-defining partition for group $G_i$, and $(S_{1,k}, \ldots, S_{n,k})$ be an arbitrary $n$-partition of $S_k$ such that $\max_{j\in[n]} c_i(S_{j,k}) \leq \mathsf{GMMS}_i$ for $1 \leq k \leq g_i$. Thus, each bundle $S_{k,j}$ for $k \in [n]$ and $j \in [g_i]$ can contain at most one large item or at most two medium items, due to the definitions of GMMS. Hence, the total number of large and medium items in $\mathcal{M}$ satisfies

$$|H_i(\mathcal{M})| + \frac{|D_i(\mathcal{M})|}{2} \leq n \cdot g_i.$$

Next, we introduce our algorithm. The algorithm runs in two phases. In the first phase, we allocate items, via Algorithm 1, to each group $G_i$, denoted by $\mathbf{B} = (B_1, \ldots, B_n)$. In the second phase, we allocate $B_i$ to agents in each group $G_i$, denoted by $B_i = (B_{i,1}, \ldots, B_{i,g_i})$.

We start with Algorithm 1, which runs in $n$ rounds. Each round $j$ consists of two parts: bag initialization and bag filling. In bag initialization, items are put in a bag $S$ according to their indices, and one will be selected in every $n$ items. Specifically, the selection adopts the following strategy:

- If there are currently $x$ items in bag $S$ and $x$ is even, the chore with index $n \cdot x + j$ will be considered next;

- If there are currently $x$ items in bag $S$ and $x$ is odd, the chore with index $n \cdot x + n + 1 - j$ will be considered;

- Assume item $e_t$ is being considered. If there is a group that regards $e_t$ as large or medium, note this group, add $e_t$ to $S$, and begin considering the next item;

- Otherwise, exit the bag initialization step.

Consequently, the items in $S$, considered medium or large by some group, contain two sequences: Items with indices

**Algorithm 1** Assign Items to Groups

**Input**: Ordered homogeneous instance $\mathcal{I} = (\mathcal{G}, \mathcal{M}, \mathbf{c})$.
**Output**: Allocation $\mathbf{B} = (B_1, \ldots, B_n)$.

1: Initialize $R \leftarrow \mathcal{M}$.
2: **for** $j$ from 1 to $n$ **do**
3:     *%Bag Initialization:*
4:     Initialize $S \leftarrow \emptyset, t \leftarrow j$.
5:     Let $k$ be the index of one group such that $G_k \in \mathcal{G}$.
6:     **while** $\exists G_i \in \mathcal{G}$ s.t. $e_t \in H_i(R) \cup D_i(R)$ **do**
7:         $S \leftarrow S \cup \{e_t\}, R \leftarrow R \setminus \{e_t\}, k \leftarrow i$.
8:         **if** $\lfloor t/n \rfloor$ is odd **then**
9:             $t \leftarrow (\lfloor t/n \rfloor + 2) \cdot n + 1 - j$.
10:        **else**
11:           $t \leftarrow (\lfloor t/n \rfloor + 1) \cdot n + j$.
12:        **end if**
13:     **end while**
14:     *%Bag Filling:*
15:     **while** $\exists G_i \in \mathcal{G}$ s.t. $c_i(S) < 1$ and $L_i(R) \neq \emptyset$ **do**
16:         $k \leftarrow i$.
17:         $e \leftarrow$ the item with the largest index in $R$.
18:         $S \leftarrow S \cup \{e\}, R \leftarrow R \setminus \{e\}$.
19:     **end while**
20:     $B_k \leftarrow S, \mathcal{G} \leftarrow \mathcal{G} \setminus \{G_k\}$.
21: **end for**

$j, 2n + j, 4n + j, \ldots$ form one, and with indices $2n + 1 - j, 4n + 1 - j, 6n + 1 - j, \ldots$ form the other.

In the bag filling step, the algorithm attempts to put more items in the bag $S$ and assign the bag to some group. It will check if there exists some group $G_i$ for which the cost of $S$ is at most 1 and there are small items unallocated. If such a group exists, the algorithm will note this group and add the smallest item (for all groups since the instance is ordered) to bag $S$. This step will be repeated until there is no group that satisfies the conditions. The algorithm will then allocate bag $S$ to the last noted group, which will leave the algorithm. Eventually, after $n$ iterations, one bag will be allocated to each group, and the algorithm terminates.

*Claim* 5. For group $G_i$, $|H_i(B_i)| + \frac{|D_i(B_i)|}{2} \leq g_i$.

*Proof.* Recall that large and medium items are added to $B_i$ in the bag initialization step. Denote $k = |H_i(\mathcal{M})| \bmod n$. Since Algorithm 1 allocates 1 item among every $n$ large and medium items into $B_i$, $|H_i(B_i)| = \lfloor \frac{|H_i(\mathcal{M})|}{n} \rfloor$ or $\lceil \frac{|H_i(\mathcal{M})|}{n} \rceil$, and $|D_i(B_i)| = \lfloor \frac{|D_i(\mathcal{M})|}{n} \rfloor$ or $\lceil \frac{|D_i(\mathcal{M})|}{n} \rceil$.

When $|H_i(B_i)| = \lfloor \frac{|H_i(\mathcal{M})|}{n} \rfloor$ or $k = 0$,

$$
\begin{aligned}
|H_i(B_i)| + \frac{|D_i(B_i)|}{2} &\leq \lfloor \frac{|H_i(\mathcal{M})|}{n} \rfloor + \frac{\lceil \frac{|D_i(\mathcal{M})|}{n} \rceil}{2} \\
&\leq \lceil \frac{|H_i(\mathcal{M})| + |D_i(\mathcal{M})|/2}{n} \rceil \leq g_i.
\end{aligned}
$$

When $|H_i(B_i)| = \lceil \frac{|H_i(\mathcal{M})|}{n} \rceil$ and $k \neq 0$, let item $e$ denote the last large item added in $B_i$ during the algorithm. According to the algorithm's selective strategy, after selecting item $e$, the next medium item (if any) is chosen after at least $2(n - k)$ items. After the $2(n - k)$ items, the number of the medium items does not exceed $(g_i - \lceil \frac{H_i(\mathcal{M})}{n} \rceil) \times 2n$. Since in bag initialization, if the algorithm selects item $e_j$, before the algorithm selects $e_{j+2n}$, only one another item is selected. In each subsequent group of $2n$ medium items, the algorithm selects 2 items to be included in $B_i$. Hence,

$$
|H_i(B_i)| + \frac{|D_i(B_i)|}{2} \leq \lceil \frac{H_i(\mathcal{M})}{n} \rceil + g_i - \lceil \frac{H_i(\mathcal{M})}{n} \rceil = g_i,
$$

which completes the proof. $\square$

*Claim* 6. All the items can be allocated by Algorithm 1.

*Proof.* Consider the last remained group is $G_i$ in the last round of the algorithm. In the bag initialization part, there is only one item unallocated in every $n$ items among group $G_i$'s large and medium items. Therefore, all the large and medium items can be added into $B_i$ during the bag initialization part. If there are some items that are small for $G_i$, for the bundle $S$ removed in every round before the last round, $c_i(S) \geq \frac{c_i(\mathcal{M})}{n}$. Therefore, for the remaining items $S'$ in the last round, $c_i(S') \leq c_i(\mathcal{M}) - (n - 1)\frac{c_i(\mathcal{M})}{n} = \frac{c_i(\mathcal{M})}{n}$. All the small items for group $G_i$ are added into $B_i$. Hence, all the items can be allocated. $\square$

*Claim* 7. Given the set of items $B_i$ computed by Algorithm 1, we can divide $B_i$ into a $g_i$-partition $(A_{i,1}, \ldots, A_{i,g_i})$ such that $c_i(A_{i,j}) \leq \frac{4}{3}\mathsf{GMMS}_i$, for any agent $a_j \in G_i$.

*Proof.* Firstly, initialize the set of bundles $A_{i,j} = \emptyset$ for all $j \in [g_i]$. We allocate one large item or two medium items to one empty bundle until all the items in $H_i(B_i) \cup D_i(B_i)$ are allocated. All large and medium items can be allocated this way by Claim 5.

After allocating the large and medium items, each agent in the group will have a cost at most $\frac{4}{3}\mathsf{GMMS}_i$, which is directly derived from the definition of large and medium items. If $L_i(B_i) = \emptyset$ then the proof is complete. Otherwise, consider $e^*$, the last item added to bag $S$(which becomes $B_i$) by Algorithm 1. We have $c_i(B_i \setminus \{e^*\}) < 1 \leq \mathsf{GMMS}_i \cdot g_i$ from Line 15 of the algorithm. For item $e \in L_i(B_i) \setminus \{e^*\}$, find some agent $a_{i,j}$ with $c_i(A_{i,j}) \leq \mathsf{GMMS}_i$ and assign $e$ to that agent: $A_{i,j} \leftarrow A_{i,j} \cup \{e\}$. Repeat this step until $L_i(B_i) \setminus \{e^*\}$ is exhausted. Note that we can always find an agent to receive $e$. Otherwise, at some point each agent in $G_i$ has cost more than $\mathsf{GMMS}_i$, which means $c_i(B_i \setminus \{e^*\}) \geq \sum_{j \in [g_i]} c_i(A_{i,j}) \geq g_i \cdot \mathsf{GMMS}_i$, asserting a contradiction. Analogously, after allocating $L_i(B_i) \setminus \{e^*\}$,

there exists some agent $a_{i,j^*}$ such that $c_i(A_{i,j^*}) \leq \mathsf{GMMS}_i$. We then assign the only remaining item $e^*$ to this agent, thus $A_{i,j^*} \leftarrow A_{i,j^*} \cup \{e^*\}$. Note that each agent who would receive a small item this way has cost at most $\mathsf{GMMS}_i$ before taking the item, so we can conclude that for each agent $a_{i,j}$ in $G_i, c_i(A_{i,j}) \leq \frac{4}{3}\mathsf{GMMS}_i$. $\qquad\square$

Based on the discussion above, Theorem 4.1 is proved.

We remark that the above algorithm is not polynomial-time, since we need to first compute GMMS values. However, by applying the techniques in (Barman & Krishnamurthy, 2020), our algorithm can run in polynomial time, and the approximation ratio will increase to $\frac{4}{3} + \epsilon$ for an arbitrarily small constant $\epsilon$.

### 4.2. Non-existence of GMMS Allocations

We finally show that an exact GMMS allocation may not exist for any number of groups, even when every group contains a single agent, which degenerates to the traditional setting of MMS fair allocation of chores with additive cost functions. It is proved in (Aziz et al., 2017; Feige et al., 2021) that when $n = 3$, an MMS allocation is not guaranteed. For $n > 3$, it remains unknown. On the other hand, Procaccia & Wang (2014) designed a series of instances for the allocation of goods, showing that an MMS allocation is not guaranteed for any $n \geq 3$. We next show how to convert these instances to chores. Note that, in this section, we restrict our focus to the case where each group contains a single agent. We use $\mathcal{I} = (\mathcal{M}, \mathcal{N}, \mathbf{v})$ to denote a goods instance, where $\mathcal{N} = \{a_1, \dots, a_n\}$ is the set of agents and $\mathbf{v} = (v_1, \dots, v_n)$ is their valuation profile. For a goods instance, agents want to get higher utilities, and their MMS is defined by the maximum, over all $n$-partitions of $\mathcal{M}$, of the minimum value of a bundle in the $n$-partition. We refer the readers to (Feige et al., 2021) for a formal definition.

**Theorem 4.4.** *For any $n \geq 3$, there is an instance that does not admit an exact MMS allocation.*

*Proof.* Let $\mathcal{I} = (\mathcal{M}, \mathcal{N}, \mathbf{v})$ be any goods instance such that (1) for every agent $a_i \in \mathcal{N}$, $\mathsf{MMS}_i = \frac{1}{n}v_i(\mathcal{M})$, and in one of her MMS-defining partitions, every bundle has the same number of goods, and (2) there is no MMS allocation. Note that the instances provided in (Procaccia & Wang, 2014) satisfy these requirements. We construct a chores instance $\mathcal{I}' = (\mathcal{N}', \mathcal{M}', \mathbf{c})$, where $\mathcal{M}' = \mathcal{M}$, $\mathcal{N}' = \mathcal{N}$, and

$$c_i(g) = v_i(\mathcal{M}) - v_i(g) \text{ for any } g \in \mathcal{M}'.$$

Denote by $\mathsf{MMS}'_i$ the MMS of agent $a_i \in \mathcal{N}'$ in $\mathcal{I}'$. Hence

$$\mathsf{MMS}'_i = \frac{c_i(\mathcal{M})}{n} = \frac{m}{n}v_i(\mathcal{M}) - \mathsf{MMS}_i = \frac{m-1}{n}v_i(\mathcal{M}).$$

For any set of chores $S$ such that $|S| \geq \frac{m}{n} + 1$, the cost of the bundle is at least

$$c_i(S) \geq (\frac{m}{n} + 1)v_i(\mathcal{M}) - v_i(S)$$
$$\geq (\frac{m}{n} + 1)v_i(\mathcal{M}) - v_i(\mathcal{M}) = \frac{m}{n}v_i(\mathcal{M}) > \mathsf{MMS}'_i.$$

Hence, if there is an MMS allocation, it must be balanced, i.e., every agent must obtain exactly $\frac{m}{n}$ chores. Since there is no MMS allocation in the goods instance $\mathcal{I}$, there must be one agent $a_i$ who gets a bundle $A_i$ such that $v_i(A_i) < \mathsf{MMS}_i$ for any allocation $\mathbf{A}$. Thus, in the chores instance, for any allocation $\mathbf{A}$, there also exists an agent $a_i$ such that

$$c_i(A_i) = \frac{m}{n}v_i(\mathcal{M}) - v_i(A_i)$$
$$> \frac{m}{n}v_i(\mathcal{M}) - \mathsf{MMS}_i = \mathsf{MMS}'_i,$$

implying that $\mathcal{I}'$ does not admit an MMS allocation. $\qquad\square$

## 5. Conclusion

In this work, we study the fair allocation problem of indivisible chores among groups of agents, where fairness is measured by group maximin share (GMMS). For the heterogeneous setting, we design a polynomial-time algorithm that returns a 2-GMMS allocation, and prove that the approximation ratio cannot be improved to better than $2 - \frac{1}{n}$. For the homogeneous setting, we improve the approximation ratio to $\frac{4}{3}$, but the best possible approximation in this case remains unknown.

Our paper uncovers many open problems and further directions. As mentioned, the best possible approximation ratio in the homogeneous setting is still unknown. In fact, for the homogeneous setting, we still do not know if it makes a difference between the case of multiple agents within each group and when every group contains a single agent. Secondly, for the heterogeneous setting, our approximation ratio is tight only when the number of groups goes large. It is an open question whether we can design better algorithms for a small number of groups. Thirdly, in this paper, we only consider identical and unrelated machines. It is interesting to extend our setting to related machines. We still do not know if the setting of related machines is strictly harder than that of identical machines.

## Acknowledgments

This work is funded by the Hong Kong SAR Research Grants Council (No. PolyU 15224823) and the Guangdong Basic and Applied Basic Research Foundation (No. 2024A1515011524).

## Impact Statement

This paper presents work whose goal is to advance the field of Machine Learning. There are many potential societal consequences of our work, none which we feel must be specifically highlighted here.

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
