# OpenReview forum: "Settling the Maximin Share Fairness for Scheduling among Groups of Machines"
_ICML.cc/2025/Conference — ICML 2025 poster_

### Official Review · Reviewer_sFkZ · 2025-03-03

**Overall Recommendation:** 3

**Summary:**

The paper studies a variant of the fair scheduling problem. There are groups of machines and tasks which need to be scheduled on these machines. The paper focuses on the fairness objective of (group) maximin share (GMMS) and shows that a 2 approximation to GMMS can be computed in polynomial time. The paper additionally improves upon a result in the literature by showing that a 4/3 approximation to GMMS exists when the agents in each group are homogenous. The paper as a whole is purely algorithmic and theoretical.

## update after rebuttal

Thank you for the rebuttal! I hope you take the improvement suggestions seriously!

**Claims And Evidence:**

The paper is purely theoretical and all claims are backed up by (for the most part understandable) proofs. All claims and evidence seems to be present and non-problematic.

**Essential References Not Discussed:**

I cannot think of any.

**Experimental Designs Or Analyses:**

N/A

**Methods And Evaluation Criteria:**

N/A due to being a purely theoretical paper.

**Other Comments Or Suggestions:**

"There are two
layers of objects, where a number of indivisible jobs (can be
viewed as data points) need to be first allocated to n groups
(viewed as clusters or super-agents). Each group controls a
number of machines (can be viewed as features or atomic
agents). Upon receiving a set of jobs, the group should
further assign these jobs to its machines with the objective
of completing these jobs as early as possible." I find this whole sentence construction to be quite confusing (in particular the "can be viewed as" parts)

"MMS fair allocation of chores and job scheduling problem" -> "MMS fair allocation of chores and the job scheduling problem"

I prefer the use of \citet for inline citations.

"does not admit better than n approximation" -> "does not admit a better than n approximation"

"Each group i contains" -> This should probably be each group G_i

I am not the biggest fan of just starting new sections with a theorem. A bit of guidance to the reader is always nicer.

I find it confusing that you call the groups large, medium, and small, but abbreviate them as H, D, L with L not being large

**Other Strengths And Weaknesses:**

In general, I like this paper. The question studied is mostly natural (one might question the relevance to the ML literature, however, ICML has opened up to more non ML papers in recent years, and has included a game theory track this year, so I believe this is okay). I however (as argued above) believe that the authors could have improved the clarity of their writing throughout the work. In particular, as this is not a paper on ML, I think to be properly appreciated by non-ML readers, I think the paper should have included more motivation for their setting and additionally guidance to the reader. I am not really a fan of not including any examples at all, just a single example after the model definition greatly improves the readability and clarity in my experience.

**Questions For Authors:**

I have no questions at the moment

**Relation To Broader Scientific Literature:**

The paper is part of the well established literature on (indivisible) fair division. Specifically, it builds upon the work of Li et al. (Neurips 2023) in studying MMS for a group scheduling problem, answers one of their open questions and improves upon their obtained bound. On a more meta-level it contributes to the understanding of MMS for non-additive fair division instances and contributes a non-trivial non-additive model in which constant factor MMS approximations can be achieved. The results and studied fairness notions are very "standard" for the fair division literature and are a generalization of commonly studied objectives in the multi-machine scheduling setting.

**Theoretical Claims:**

I checked the proofs, however, I could not follow the proof of Theorem 3.2 fully. For the most part the proofs are understandable, however, I believe that the authors could have put more work into making them easier to digest. In particular, I would have been a fan of (even more) visualization in the proofs. For instance, the delegation of the illustration in Theorem 3.1 does not really help the proof (and I would have preferred it to be reversed). I think in general, the paper would have been nicer, if the authors maybe removed one of the proofs and instead focused more on extending/explaining the other two.

---

> ### Author Rebuttal · Authors · 2025-03-31
>
> We thank the reviewer for the appreciation of the paper and the detailed and constructive comments. We will carefully revise the paper following these suggestions.
>
>
> *Comment*: On the theoretical proofs.
>
> *Response*: We thank the reviewer for the suggestion of restructuring the proofs.
> We will focus more on explaining and visualizing the proof ideas in the main body (including adding examples and figures), possibly putting some proofs in the appendix.
>
>
> *Comment:* On the motivations and examples.
>
> *Response*: We thank the reviewer for the suggestion of including more motivation in the introduction and one example after the definition. We will follow this suggestion to revise the paper. In particular, we will expand the motivating example in the introduction to motivate our setting (e.g., a group’s workday can conclude when all members complete their assigned duties) and add one concrete example after the definition of GMMS. We will also include a figure to illustrate the partition and the notations in GMMS.
>
>
> *Comment*: On the other comments.
>
> *Response*:
>
> - We will rephrase the sentence "There are two layers of objects, ..., with the objective of completing these jobs as early as possible" and the corresponding paragraph. Basically, we want to explain how the items are first allocated to groups and how the group will further distribute the items among its members. We will avoid using ambiguous words like "can be viewed as".
>
> - We will use \citet for inline citations.
>
> - We will add guidance at the start of each (sub)section.
>
> - Regarding the abbreviations of large, medium, and small groups, we used H and L to represent high and low costs. We understand the ambiguity and will revise this part accordingly.
>
> - We will also address the other grammar issues and carefully revise the writing in the revised version.

---

### Official Review · Reviewer_u6td · 2025-03-07

**Overall Recommendation:** 4

**Summary:**

This paper addresses the maximin share (MMS) fairness problem in the context of job scheduling among groups of machines. . The study paper builds upon the work of Li et al. (NeurIPS 2023), which considered MMS fairness for groups of identical or related machines but left open the case where machines within a group are unrelated. The authors provide a polynomial-time algorithm that computes a 2-approximate MMS allocation in the case of heterogeneous machines using linear programming techniques. The paper proves that no algorithm can achieve better than a (2 - 1/n)-approximate MMS allocation. When machines within each group are identical, the paper improves the approximation ratio from 2 to 4/3.

**Claims And Evidence:**

Yes

**Essential References Not Discussed:**

N/A

**Experimental Designs Or Analyses:**

N/A

**Methods And Evaluation Criteria:**

Yes

**Other Comments Or Suggestions:**

No

**Other Strengths And Weaknesses:**

Strengths:
-The paper is well-written and closes an open question that was left from a previous work.

-The results are clean, with clear proofs and tight bounds.

-Also, nice linear programming techniques are used that make the proofs quite novel.

-The problem is well-motivated  and well-defined.

**Questions For Authors:**

Right column, line 118: Do you mean all k-partitions of S?

Right column, line 127: Is this correct: "of her bundle with the other n−1 groups."?
Also, in the equation below MMS(Sk,n,ci,k) is n correct?

**Relation To Broader Scientific Literature:**

Very related

**Theoretical Claims:**

I did not check thoroughly all the proofs but I checked the high level ideas and they seem correct.

---

> ### Author Rebuttal · Authors · 2025-03-31
>
> We thank the reviewer for the positive feedback and the constructive comments. In the following, we answer your questions. We will carefully address these questions in the revised paper.
>
> *Question*: On line 118, all $k$-partitions of $S$.
>
> *Answer*: Yes, it is a typo, which should be all $k$-partitions of $S$. We will fix this in the revised version.
>
> *Question*: On line 127, $n-1$ groups and $n$ in $MMS(S_k,n,c_{i,k})$.
>
> *Answer*: The two statements are indeed correct but might be a little unclear. To define $GMMS_i$ for group $G_i$, we first partition all items into $g_i$ bundles, where the $k$-th bundle $S_k$ is further allocated to $n$ (imaginary) agents, and each of them has the same valuation as $a_{i,k}$'s (like the classic definition of MMS with no groups).  We will carefully refine the statements and clarify this definition in the revised version. If it helps, we will include a figure to illustrate the partition and these notations.

---

### Official Review · Reviewer_MsHm · 2025-03-13

**Overall Recommendation:** 4

**Summary:**

The paper discusses the problem of group maximin share job allocation, where groups need to be assigned sets of jobs which are distributed to machines within each group to ensure the minimum largest makespan within that group. In the heterogenous setting, where machines within a group can have different cost functions, the authors show a lower bound of 2-1/n approximation ratio using a difficult example, and an upper bound of 2-MMS  approximation ratio by constructing a polynomial time algorithm for constructing an allocation. Further, in the homogenous case, they reduce the previous upper bound of 2-MMS to 4/3MMS (when the Group MMS allocations are known) and 4/3MMS +$\epsilon$ relaxation in polynomial time.

Post-rebuttal comment:
The author's rebuttal addresses some of my concerns, and as long as they fix the typos and minor errors, I am in favor of accepting this paper.

**Claims And Evidence:**

The paper presents detailed proofs of the three claims.

**Essential References Not Discussed:**

The paper has good related work

**Experimental Designs Or Analyses:**

NA

**Methods And Evaluation Criteria:**

Yes

**Other Comments Or Suggestions:**

NA

**Other Strengths And Weaknesses:**

One general comment I have about the paper is the slight confusion the term Maximin Share induces. In typical fair division literature, MMS refers to maximizing the worst-off agent's utility, but in this case, the problem is minimizing the maximum makespan, bringing it closer to a minimax setting.

There is also some inconsistency within the paper with how approximation ratios are presented. In most of the paper, approximation ratios larger than 1 are used, but in the related work, some of the values mentioned are below 1 (suggesting they are better than optimal?). This seems like a mistake, and I would encourage the authors to review the text and make this consistent.

This is also present in section 4.1. The title mentions a 3/4 GMMS algorithm, while the theorem states 4/3 GMMS.


Another note for the introduction: in the Introduction section, it seems like the message isn't clearly conveyed. It talks about indivisble jobs vs data points, machines, features and atomic agents. It is talking about too many different kinds of things, which sends a confusing message. Since the paper deals with the job scheduling problem, sticking with one single narrative would be better for readers trying to understand the contributions of the paper.

**Questions For Authors:**

No questions

**Relation To Broader Scientific Literature:**

The paper pushes the frontier of group job allocation, presenting new algorithms and improved upper and lower bounds for the problem setting of improving maximin share fairness. The designed counterexample proves a good lower bound for heterogenous utilities, which has not been seen before.

**Theoretical Claims:**

I checked the proofs for the three theoretical claims, and was convinced of their correctness. I was not able to completely follow the final proof for Theorem 4.1 (4/3 GMMS allocation upper bound), but I can see how it holds.

---

> ### Author Rebuttal · Authors · 2025-03-31
>
> We thank the reviewer for the appreciation of the results and the insightful comments. We will carefully review the paper to refine the presentation, including the introduction and the technical proofs.  Below, we address your specific comments.
>
> *Comment*: On the term of Maximin Share.
>
> *Response*: The reviewer is correct that MMS should actually refer to the "minimax share".
> However, to be consistent with the allocation of goods (i.e., maximizing the worst-off agent's utility), the use of "maximin share" forms a convention in the literature of allocating chores (i.e., minimizing the maximum makespan). This is partially because the very first several works on this topic used "non-positive utilities" instead of "non-negative costs" to describe the valuation functions. Recent works on chores found it is more convenient to use non-negative costs, and maximin share is used for consistency. We will carefully explain the notations in the revised version and would be happy to make changes based on the reviewer's advice.
>
> *Comment*: On the range of the approximation ratio.
>
> *Response*:  We thank the reviewer for pointing out this issue. It is a typo in the title of Section 4.1, where the approximation ratio should be 4/3. To define the approximation ratios, we adopt the following rule.
>
> - For the problem of goods (corresponding to maximization objectives), the approximation ratio $\alpha$ is defined to be smaller than 1, and $\alpha$-approximation means the algorithm can guarantee at least $\alpha$ fraction of the optimal utility;
>
> - For the problem of chores (corresponding to minimization objectives), the approximation ratio $\alpha$ is defined to be greater than 1, and $\alpha$-approximation means the algorithm can guarantee no more than $\alpha$ times the optimal cost.
>
> We will carefully review the paper and ensure consistency in approximation ratios.
>
> *Comment*: On the presentation of the introduction.
>
> *Response*: We thank the reviewer for the suggestion of sticking with the job scheduling problem.
> We will reorganize the introduction and avoid talking about different kinds of things.

---

### Official Review · Reviewer_x4k3 · 2025-03-18

**Overall Recommendation:** 3

**Summary:**

This paper considers a fair resource allocation problem called Group Maximin Share (GMMS). There are two layers of allocation: at the first layer, there are $m$ items to be allocated to $n$ groups. Then, once items are allocated to each group $G_i$, they are further allocated to the $g_i$ agents in $G_i$. Each agent has a cost for each item, and the agent's cost on the allocation they receive is the makespan of the items; the cost of the group is the maximum makespan among the agents. The maximin share $\textsf{GMMS}_i$ of the group $G_i$ is computed as follows (Lemma 2.3) in the paper: consider all possible partitions of $m$ items into $n$ parts. Consider the part $S$ that, when allocated to group $i$, has the worst cost. Here, cost is referring to the second layer, where it is the minimum maximum makespan among all possible allocations of $S$ to the $g_i$ agents in $G_i$. So the problem can be written as two successive min max problems.

The fairness goal is to find an allocation (at both layers) such that the cost of any group (computed as the maximum of the agents' makespans) is at most $\alpha \cdot \textsf{GMMS}_i$. The authors show that for the heterogeneous case (agent's costs within a group are potentially different), there is an algorithm yielding $\alpha = 2$, and that this is asymptotically tight ($2-1/n$). They also show that for homogeneous costs, there is an algorithm yielding $\alpha = 4/3$.

## update after rebuttal:
As my concerns were mainly about clarity of writing, I maintain my evaluation.

**Claims And Evidence:**

Full proofs are provided.

**Essential References Not Discussed:**

The most relevant paper, Li et. al. (2023) is discussed, along with a number of references on MMS more broadly. I did not notice any important omissions for understanding the paper.

**Experimental Designs Or Analyses:**

Not applicable.

**Methods And Evaluation Criteria:**

No experiments are provided; this is a pure theory paper.

**Other Comments Or Suggestions:**

- pg.5: referring to $k$th copy of $G_i$ is confusing language
- Sometimes ratios are listed as the reciprocal of what they should be.
- In Theorem 4.4, it would be clearer to say something like an exact allocation.
- It would be good to define what related machines means in this context

**Other Strengths And Weaknesses:**

This paper contributes a worthy contribution of nearly matching upper and lower bounds for the heterogeneous case. The idea behind the proof of the 2-GMMS is nice: the authors compute a fractional solution to the LP, and then construct an auxiliary graph where edges are labelled with the fractional allocations. Then the solution is rounded to an integral one by sending allocations through the graph. While the 4/3-GMMS result for the homogeneous case is not proven to be tight, it is a notable improvement over previous work.

The main weakness of the paper is that the writing is unclear, particularly in the first half and with respect to both related work and the model at hand. Several problems are discussed without defining the relevant terminology, which makes it difficult to contextualize the present results. There is also an abuse of language, as approximation ratio is used to describe the fairness factor $\alpha$, but this is not how approximation ratio is used in classic approximation algorithms (while the term is also used in the regular sense to discuss some related works, as I understand).

**Questions For Authors:**

n/a

**Relation To Broader Scientific Literature:**

The most closely related literature is the work of Li et. al. (2023), where the problem was introduced. They showed a $O(\min\{n, \log m\})$-GMMS for the heterogeneous case and a 2-GMMS for the homogeneous case. The present work improves the first factor to 2 and the second factor to 4/3.

**Theoretical Claims:**

I did not check proofs in any detailed manner.

---

> ### Author Rebuttal · Authors · 2025-03-31
>
> We appreciate the reviewer's supportive review and constructive suggestions. We will carefully address the typos and polish the presentation.
>
> *Comment*: On the terminologies in related work and the model.
>
> *Response*: We thank the reviewer for pointing out this issue.
> We will thoroughly examine the paper and improve the writing.
> Specifically, we will focus on providing clear definitions for the terms used and expanding on the discussed models (e.g., related machines).
>
>
> *Comment*: On the definition of approximation ratios.
>
> *Response*: We thank the reviewer for raising this issue, and we agree that the approximation ratio is classically used in optimization problems to measure the multiplicative factor between efficient and optimal solutions.
> However, in the literature of MMS fair allocation of indivisible items, it becomes the convention to use the term "approximation ratio" to describe the best possible achievable factor of MMS.
> We thought it might be better to be consistent with the literature when preparing the submission, but we also understand the potential ambiguity.
> Perhaps "approximation factor" and "competitive ratio" make more sense.
> We are open to adjusting this terminology based on the reviewer's advice.
>
>
> *Comment*: On pg.5, the kth copy of $G_i$.
>
> *Response*: $k$ refers to the $k$-th bundle in the $n$-partition of group $G_i$. We will rephrase this sentence to make it clear.
>
> *Comment*: Sometimes ratios are listed as the reciprocal.
>
> *Response*: We will check the ratios carefully and make them consistent and accurate.
>
> *Comment*: On Theorem 4.4.
>
> *Response*: Thanks for the suggestion. We will mention "exact allocation" in the description of the theorem.
>
> *Comment*: On the definition of related machines.
>
> *Response*: Thanks for the suggestion. We will include the definition of related machines the first time we mention it in the paper.
> We will also review other places to ensure the clarity of terminologies and models.

---

### Decision · Program_Chairs · 2025-05-01

**Decision:**

Accept (poster)

**Comment:**

This submission addresses a two-layer fair allocation (scheduling) problem referred to as Group Maximin Share (GMMS). In the first layer, a set of indivisible items (jobs) is distributed among multiple groups. Within each group, there is a second layer of allocation to the machines (or “agents”) in that group. Each agent’s cost is determined by the makespan of the items it receives, and the group’s cost is defined as the maximum makespan among its agents. The paper seeks fair allocations that achieve a bounded approximation with respect to each group’s GMMS value.

The paper gives a 2-approximation in the hetrogeneous case and a 4/3 approximation in the homogeneous case.

The reviews pointed out strong points in the paper. (1) It solves a noteworthy open question from prior work (Li et al., NeurIPS 2023). Establishes new results with rigorous theoretical proofs and nearly tight bounds. Employs a linear programming approach and rounding ideas that are both interesting and non-trivial. Offers a meaningful theoretical contribution to the fair division (MMS/GMMS) literature.

Reviewers pointed out that the presentation could be

Despite the minor concerns around presentation, all reviewers agree that the paper’s contributions are significant and that it resolves meaningful open questions with strong theoretical results.